# Deleterious Effect of Ultraviolet Radiation on *Glossogobius giuris:* A Short Experimental Study

**Azhagu Raj Ramakrishnan** [1,*], **Krishna Kumar** [2], **Palavesam Arunachalam** [2], **Muthupandi Sankar** [3], **Prathap Selvaraj** [3] and **Sohan Jheeta** [4,*]

1 Department of Zoology, St. Xavier's College (Autonomous), Palayamkottai 627002, India
2 Department of Animal Science, Manonmaniam Sundaranar University, Tirunelveli 627012, India; kumarkrishk89@gmail.com (K.K.); plavesh06@gmail.com (P.A.)
3 Department of Physics, Loyola College, Affiliated to University of Madras, Chennai 600034, India; muthupandisankar@gmail.com (M.S.); mic.prathap@gmail.com (P.S.)
4 Network of Researchers on the Chemical Evolution of Life (NoRCEL), Leeds LS7 3RB, UK
* Correspondence: drazhaguraj@gmail.com (A.R.R.); sohan@sohanjheeta.com (S.J.)

**Abstract:** Ultraviolet (UV) radiation is a part of the spectrum of electromagnetic radiation emitted by the Sun. The present study was conducted to examine the deleterious effects of UV radiation on the stratum corneum of fish—namely, *Glossogobius giuris*. In this study, healthy living specimens of *G. giuris* species weighing (1.20 g) and length (4.06 cm) were collected from Thandavarayankulam lake, Srivaikuntam Taluk, Thoothukudi District, Tamil Nadu. They were transported to the laboratory in well-aerated containers. During the experiment, the fish *G. giuris* (n = 6) was introduced into the UV Chamber (UVA and UVB) for one hour. After that, experimental fishes were collected from the UV Chamber were dissected for histological and biochemical studies using standard methodology. The short-term exposure of UVA and UVB rays on freshwater *G. giuris* muscle tissue showed marked degeneration of the epithelium, the disappearance of striations, thickened septal wall, broken fibre, and the disappearance of striation, followed by branchial arterial rupture. It was also determined that carbohydrate, protein, and lipid contents of the muscle tissue were significantly reduced. This study confirmed the destructive effects of UV radiation on the stratum corneum of fish *G. giuris*. The ultrastructural and biochemical changes occur depending largely on the energy of the UV rays; in this case, the UVB radiation, with higher destructive energy (4.4 eV), had a greater detrimental effect on the muscles of *G. giuris* than UVA, with its energy level of 3.9 eV.

**Keywords:** *Glossogobius giuris*; ultraviolet rays; histology; histopathology; biochemical analysis



## 1. Introduction

Aquatic ecosystems are vital components of the Earth's biosphere, as over 70% of the Earth's surface is covered by water. The ecosystems include both freshwater and saltwater habitats as exemplified by lakes, ponds, lagoons, rivers, streams, wetlands, and swamps, as well as marine habitats such as oceans, seas, intertidal zones, reefs, sea-beds, etc. The inhabitants of these ecosystems are routinely exposed to solar ultraviolet (UV) rays in the form of UVA and UVB radiation, and in the process, the young (e.g., eggs and larvae) of aquatic dwellers are subjected to a daily dose of UV rays [1–3]. Further, in relation to UVA and UVB rays, an increasing number of weather stations and networks have reported rising levels of solar radiation impinging onto the surface of aquatic systems [4–9]. Recent studies have shown that UVA and UVB radiation does penetrate ecosystems to various depths; therefore, these UV rays affect both the entire aquatic ecosystems and the aquatic species that reside within such ecosystems. Thus, the UV rays directly affect marine and freshwater ecosystems, including the primary producers and carnivores within the food chain [1,10,11].

Fish scales and epidermis are external surface barriers that are in direct contact with the aquatic environment. Such barriers protect aquatic organisms from adverse different elements in the environment, for example, effluents discharged by factories and farms, household products (such as microparticles in face-scrubbing products), chemical leakage from laboratories, indiscriminate disposal of both low and high levels of radioactive wastes, environmental pollutants from natural sources such as acidic rain, seeping of radon gas from the interior of the Earth's mantle, and solar UV radiation. Thus, the survival of marine life could be an indicator of both water quality and the health of marine life, in particular, that of freshwater. Our motivation for this research was that a large number of communities rely upon both freshwater and healthy fish stocks. In order to study the health of an ecosystem, we investigated the effect of UV light under laboratory conditions on *Glossogobius giuris,* a freshwater fish.

## 2. UV Light and Motivation

The Sun generates three types of UV radiations—namely, C, B, and A rays. The first, UVC (200–280 nm), is highly harmful; it is generally absorbed by the stratospheric ozone and oxygen, which means it does not reach the surface of the Earth; by contrast, both UVB (280–320) and UVA (320–400 nm) can reach the Earth's surface (Table 1). The former is classified as moderately energetic and ergo less destructive to life forms on the Earth when compared with UVC rays, and the latter is mildly energetic and is the least harmful.

**Table 1.** A summarised list of the three types of UV radiation and their effects on humans. We also converted the wavelength (nm) to electronvolts (eV) for easier visualisation of energies involved. The higher the electronvolts number, the greater the energy delivered to the human skin.

| Electromagnetic Radiation Bands | Wavelength (nm) | Energy in Electronvolts (eV) | Lethality | Effects Due to Exposure to the UV Radiation |
|---|---|---|---|---|
| UVC | 200–280 | 6.2–4.4 | Highly energetic; absorbed by ozone and oxygen. It does not reach the Earth's surface. | It causes skin burns as well as cancer, leading to a painful death. |
| UVB | 280–315 | 4.4–3.9 | Moderately energetic; less destructive | Overexposure may cause sunburn as well as some forms of skin cancer. It destroys vitamin A in the skin. |
| UVA | 320–400 | 3.9–3.0 | Mildly energetic; least destructive | Relatively safe but also destroys vitamin A in the skin; this may have side effects. |

The freshwater fish *G giuris* is an edible fish, commonly known as tank gopi, (or Uluvai in Tamil). It is highly widespread in all freshwater lakes and ponds in Tamil Nadu state, India. They are also distributed in Bangladesh, Malaysia, China, and East Africa. *G. giuris,* along with other marine creatures, are typically affected by both UVB and UVA radiation from the Sun. These effects are noticeable at all stages of a fish's life—from the embryo, larvae, juveniles, and through to adults; therefore, we used this fish to determine (a) the health of the fresh body of water and (b) to maintain healthy population so as to secure the future food-stock levels. In this paper, we report the results of the effects of UVB and UVA under laboratory conditions.

## 3. Materials and Methods

In this section, we discuss the following issues in the specific order indicated: ethical issues, collection and maintenance of freshwater fish, and application of experimental radiation.

### 3.1. Ethical Issues

Initially, we consulted the following website [12]. Further, this research was carried out with the full knowledge of the Department of Zoology, St. Xavier's College (Autonomous),

Palayamkottai, India, and was supervised by Dr. Azhagu R.R—the author and leader of this project. We performed full calculations pertaining to the number of photons impinging on the surface of the water within the beaker (fluence, photons/cm$^2$) in which a single fish was placed.

The following parameters were given: operating voltage of the UV lamp, 240 V; power usage by UVA, 6 watts and by UVB, 4 watts.

$$I = P/V \text{ (where I = current; P = power; V = voltage)} \tag{1}$$

$$N = I/q \text{ (where N = total number photons generated per second, s; q = } 1 \times 10^{-19} \text{ C)} \tag{2}$$

$$N = (Pq)/V \tag{3}$$

### 3.2. Collection and Maintenance of Freshwater Fish

Healthy living specimens of *G. giuris* species weighing approximately 1.20 g and with a length of 4.07 cm were collected from Thandavarayankulam lake, situated at Srivaikuntam Taluk, Thoothukudi District, Tamil Nadu; the fish were transported to the laboratory in a well-aerated fish tank to prevent hyperactivity and thus stress to the fish, as well as protect against physical injuries (Figure 1).

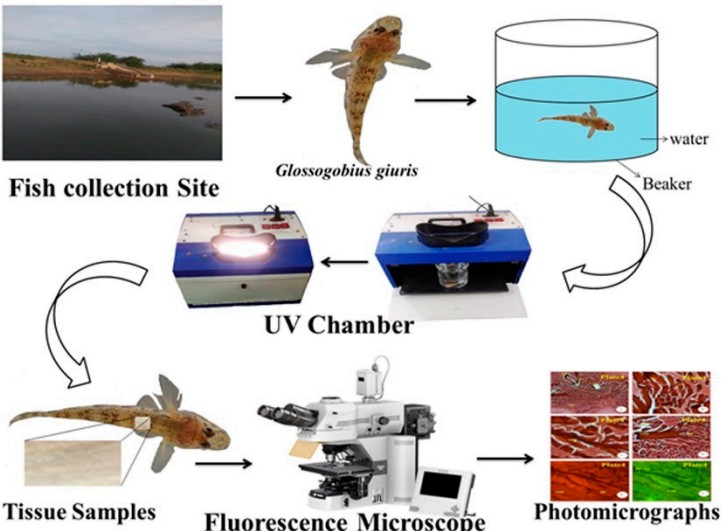

**Figure 1.** The flow diagram for the transportation of *G. giuris* from its capture to the laboratory. A single fish was transported in a large fish tank per journey in order to make sure that no stress was suffered by any fish.

The fish were maintained in a well-aerated freshwater fish tank with the following parameters: 1000 L capacity; pH 7.4; water temperature of 26 °C; photoperiod 12:12 (light:dark) for acclimatisation.

### 3.3. Experimental Setup

In this study, the control fish (n = 50) were kept inside the 1000 L departmental fish tank. During the experiment, the *G. giuris* sample (a single fish in 500 mL glass beaker filled with fresh water up to 250 mL, with water pH 7.4, the water temperature at 28 °C, and room temperature at 35 °C) was introduced to the UV chamber (Deep Vision Instruments, Chennai, India) with UVA at 320 nm (3.9 eV) and UVB at 280 nm (4.4 eV) for periods of one hour (Figure 1). This experiment was repeated six times, deploying a total of 12 fish, including the controls.

*3.4. The Ethical Issues, Applied Experimental Radiation, and Well-Being of Fish*

These experiments were carried out in accordance with the guidelines of Manonmaniam Sundaranar University's Department of Animal Science (India) (MSU/DAS/EC/2016 dated 10 March 2016) and consulted on 10 March 2017, as well as a paper entitled 'Ethical considerations for field research on fishes' [13] for the care of experimental fishes. The safety and ethical protocols were followed during the experiments, and asphyxia or mortality of fish was not encountered. Afterwards, the experimental fishes were collected from the UV chamber and were then dissected, and the muscles were taken for histological and biochemical studies.

*3.5. Preparation of Fish for Dissection*

The experimental fish *G. giuris* was anaesthetised by clove oil (Eugenol) and ethanol (95%), purchased from Sigma Aldrich, India. The concentration of anaesthesia solutions used for this study comprised eugenol and ethanol in a 1:10 ratio (100 mg/mL). After that, the experimental fish muscle tissues were taken for histological studies [14].

*3.6. Preparation for Histological Examination*

Histological observations of the muscle tissue of the control and the UVA- and UVB-radiation-exposed freshwater fish *G. giuris* were taken, and any alterations in muscle tissue were recorded. The muscle tissue samples of the fish were sliced and fixed using 50% formaldehyde for 24 h. To remove the formaldehyde, the tissues were gently washed with distilled water, and dehydration was carried out in 60%, 70%, 80%, 90%, and 100% of isopropanol for 1 h each. After cleaning with Xylene, the tissues were incubated in paraffin wax for wax impregnation, and 6 μm sections were made using a hand rotatory microtome. Then, the wax was removed, and the tissues were stained with haematoxylin and eosin. The tissue samples were mounted with DPX (distyrene, plasticiser, and xylene). The slides were observed under a Nikon 80i fluorescence microscope [15,16].

*3.7. Biochemical Analysis*

Both the control, and the experimental exposed (UVB and UVA) *G. giuris* muscle tissue proteins [17], carbohydrates [18], and lipids [19] were all analysed using UV–vis spectrophotometer (TECOMP 8500, Hong Kong).

*3.8. Statistical Analyses*

The mean and variance for muscle tissue macromolecules such as protein, carbohydrate, and lipid of both the control and (either UVB or UVA) experimental-exposed *G. giuris* were calculated using MS Excel.

## 4. Results and Discussion

*4.1. Histopathology*

The distinct straight and smooth striation features, shown in Figure 2, were observed in the muscles of the control fish. The myotomes are compact, with equally spaced muscle bundles. Muscle fibres are elongated and compact, with a number of lipid vacuoles. The qualitative septal walls between the muscle bundles appear to be thin, and the pigmentation in Figure 2 is a prominent feature.

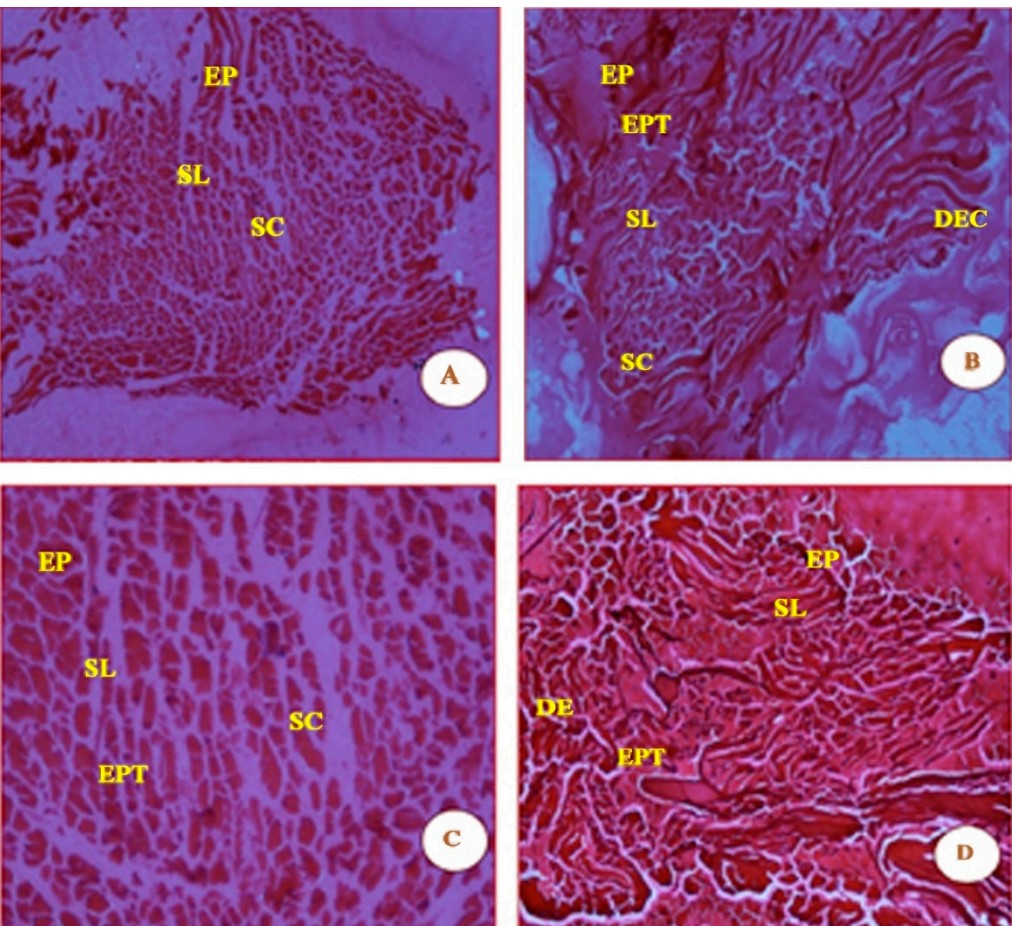

**Figure 2.** Micrographs (A, B, C and D) are shown of control fish *G. giuris* muscle sections stained with haematoxylin and eosin. The micrographs also depict clearly defined features in the four panels which are labelled as follows: epidermis (EP), stratum laxum (SL), stratum compactum (SC), epithelial tissue (EPT), dermis (DE), and dense epidermal cells (DEC). The magnification of each panel (**A–D**) was 4×, 10×, 20×, and 40×, respectively.

As seen in Figure 3, the short time exposure of UVA rays in freshwater fish *G. giuris* muscles tissue shows marked degeneration of the epithelium, the disappearance of striations, thickened septal wall, and broken fibres, followed by branchial arterial rupture, when compared with the control fish in Figure 2. Further, degenerative changes were observed in fish muscles (Figure 3) when it was irradiated with a photon fluence of $6.3 \times 10^{14}$ photons/cm$^2$ that had an energy of 3.9 eV. Fibres were withered and broken as a result of the striations having been 'dissolved'. In addition, the observed lipid vacuolation and visible pigmentation, as in Figure 2, seem to have disappeared.

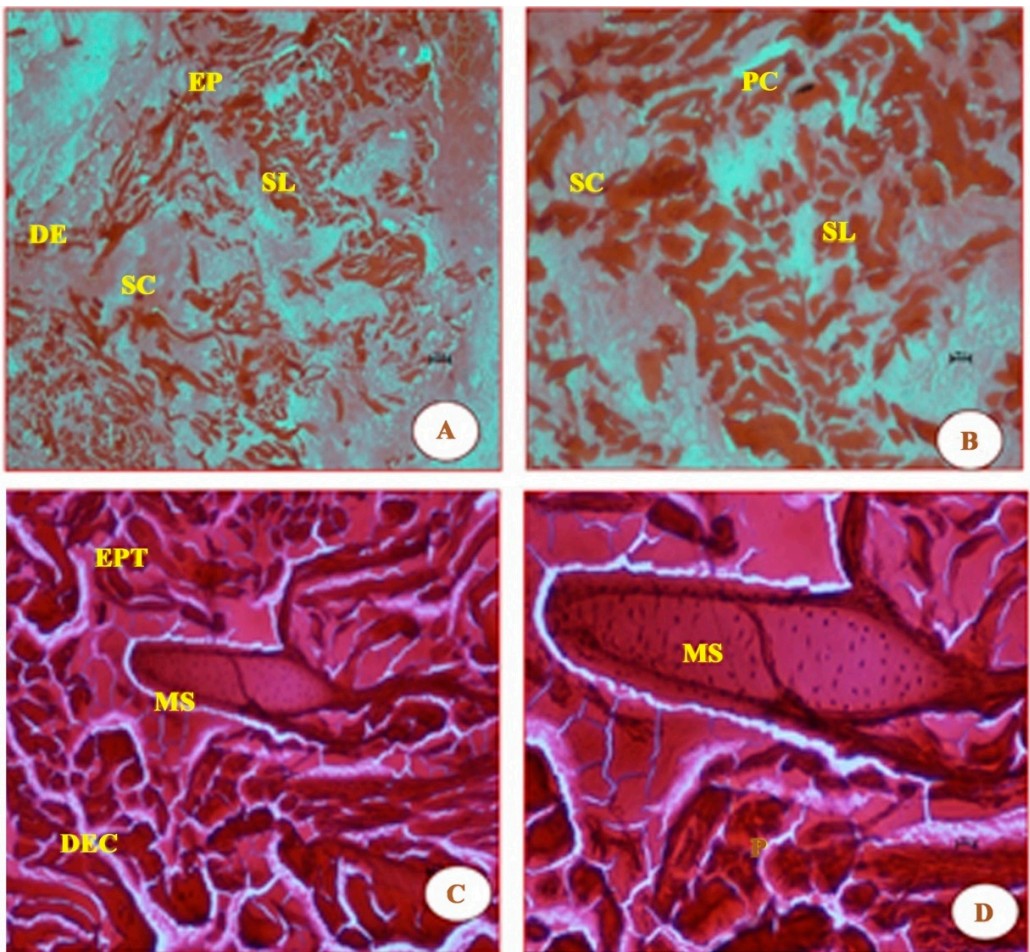

**Figure 3.** Essentially, the same features as Figure 2 are shown, which are as follows: epidermis (EP), stratum laxum (SL), stratum compactum (SC), epithelial tissue (EPT), dermis (DE), dense epidermal cells (DEC); they were also all stained with haematoxylin and eosin. The micrographs noted in these panels are due to the effect of UVA (λ = 320 nm) radiation. Panels (**A**,**B**) in Figure 3 show the difference in damage to the various tissues when compared to those in Figure 2, i.e., the tissues in Figure 2 panels A and B are smooth and undamaged. The additional new features observed in panels (**C**,**D**) are mucus cells (MS)—the MS featured in the latter two panels became visible due to samples being magnified at 20× and 40×, respectively. Panels (**C**,**D**), in particular, also show cells to be enlarged and rough, compared with those in Figure 2, which are small and smooth.

Similar to Figure 3, the micrographs in Figure 4 shows that the damage to muscle fibres is more prominent, as the UVB carries higher levels of energy (4.4 eV). It is probably because of this energy that new features such as the thickened septal wall (TSW), the squamous epithelium (SE), and squamous cells (SCs) are observed in the micrographs of Figure 4.

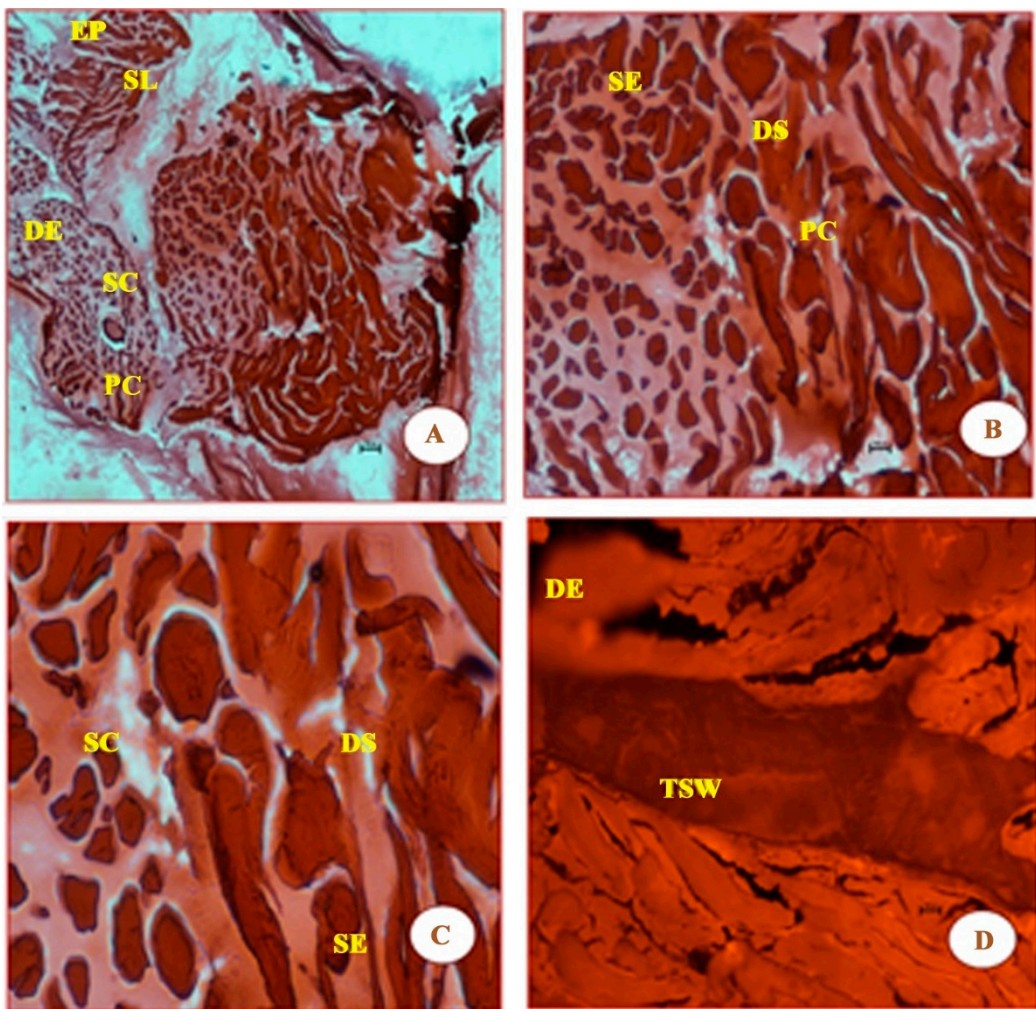

**Figure 4.** Panels (**A**–**D**) clearly show that there is more visible damage than shown in the four panels in Figure 3. In addition to epidermis (EP), stratum laxum (SL), stratum compactum (SC), and dermis epithelium (DE), which are found in Figures 2 and 3, Figure 4 also shows the following features: thickened septal wall (TSW), squamous epithelium (SE), squamous cells (SC), and pigment cells (PC). The latter four features were probably due to the strong energy of photons of UVB ($\lambda$ = 280 nm) radiation. Again, the cells in these panels appear to be both large and rough, compared with those from Figure 2. The magnification of the cells in the four panels are 4×, 10×, 20×, and 40×, respectively.

It can be observed that there are 5.25 (6.3/1.2, Table 2) more photons of UVA than those of UVB, but there is considerably more qualitative damage inflicted by the UVB, because they carried a greater amount of energy (3.9 versus 4.4 eV) and yet more fluence of photons seemingly has no effect ($6.3 \times 10^{14}$ photons/cm$^2$ for UVA versus $1.2 \times 10^{14}$ photons/cm$^2$ for UVB (Table 2)). Naturally, these results show that the shorter the wavelength ($\lambda$) is, the higher the energy it carries. It should be noted that the Sun's rays will have more severe effects on all living entities on the Earth, including aquatic ecosystems, as the total power it delivers onto the Earth's surface is 1413 W/m$^2$. Compare this with the power delivered by UVA and UVB ray lamps at 693 and 462 W/m$^2$ respectively, (under laboratory conditions, Table 2); noting that power is another way of measuring energy. This means UV radiation will definitely damage fish eggs, larvae, fish-fry, and fish fingerlings, leading to egg hatchability and increased egg incubation duration, which, in turn, results in increased damage to DNA. Such damaging of DNA will affect developmental abnormalities and cause oxidative stress, lipidosis, epidermis necrosis, and an increase in the metabolic rate [20]. It has further been shown that radiation also damages the structure and morphology of

the epidermis, causing degeneration of the epithelium, the disappearance of striations, thickening of the septal wall, and broken fibres [21], as previously mentioned. Further studies on African catfish *C. gariepinus* [22,23] also demonstrate adverse impacts by these radiations on the skin, macromolecules including proteins and lipids, enzymes (e.g., lactate dehydrogenase, glutathione reductase, and catalase), and liver function. Finally, UV radiation damage is dose-dependent, as well as stage- and species-specific [20,24].

**Table 2.** A summarised list of the fluence (photons/cm$^2$) on the surface of water within the beaker containing a single fish (fluence is number of photons impinging on the surface of water). We also calculated the power of both the UVA (693 W/m$^2$) and UVB (462 W/m$^2$), respectively, and then compared them with that of solar power delivered (1413 W/m$^2$) on the surface of the Earth.

| Electromagnetic Radiation Bands | * Total Number of Effective Photons Generated | ** Total Number of Effective Photons/cm$^2$ Generated | *** Total Power Delivered by the Lamp W/m$^2$ |
|---|---|---|---|
| | UVA | | |
| Number of photons (P = 6 W; voltage at 240 V) UVA lamp wavelength at 320 nm | $5.46 \times 10^{16}$ | $6.3 \times 10^{14}$ | 693 |
| | 3.9 eV **** | | |
| | UVB | | |
| Number of photons (P = 4 W; voltage at 240 V) UVB lamp wavelength at 280 nm | $1.04 \times 10^{16}$ | $1.20 \times 10^{14}$ | 462 |
| | 4.4 eV | | |

* The effectiveness of the UV lamp to generate photons was rated at 35%. ** Surface area of the water within the 500 mL beaker was 86.60 cm$^2$—this was necessary for determining the number of photons impinging on the surface. The volume of water within the beaker was 649.43 cm$^3$, as the beaker was 7.5 cm tall with a diameter of 10.5 cm; this is important for the well-being of the fish, e.g., by ensuring sufficient amount of oxygen within water during experimentation. *** An average global power delivered by the Sun's radiation is 1413 W/m$^2$. **** Energy of photons generated by a particular wavelength is given in electronvolts.

### 4.2. Effects of UVA and UVB Rays on the Tissues of G. giuris

The carbohydrate, protein, and lipid contents in muscle tissue were analysed in freshwater fish *G. giuris* at room temperature (control), exposed to UVA and UVB rays for an hour. The changes in total carbohydrate, protein, and lipid content in the muscle tissues were studied with respect to UV radiation, and the results are summarised in Table 3.

**Table 3.** The biochemistry of muscle tissue altered when exposed to UV rays—for example, compared with the control, the carbohydrate, protein, and lipid contents are drastically affected after 1 h of exposure to both UVA and UVB rays. For instance, carbohydrate in control sample is at 1.321 mg/100 mg of wet tissue, but when irradiated with UVA at 3.9 eV, the carbohydrate content is 0.882 mg/100 mg wet tissue, and when irradiated with UVB, the carbohydrate figures are even less, i.e., 0.311 mg/100 mg wet tissue. This trend was repeated with protein and lipid content. Biochemically, UV radiation with more energetic photons (e.g., UVB) had greater effects on fish tissues.

| Condition | Duration of Exposure | Biochemical Composition | | |
|---|---|---|---|---|
| | | Carbohydrate (mg/100 mg Wet Tissue) | Protein (mg/100 mg Wet Tissue) | Lipid (mg/100 mg Wet Tissue) |
| Control Visible light | 1 h | $1.321 \pm 0.018$ | $2.213 \pm 0.245$ | $0.211 \pm 0.020$ |
| UVA | 1 h | $0.811 \pm 0.025$ | $1.441 \pm 0.1993$ | $0.196 \pm 0.009$ |
| UVB | 1 h | $0.311 \pm 0.037$ | $1.121 \pm 0.0915$ | $0.088 \pm 0.007$ |

The muscle tissue carbohydrate content of *G. giuris* reduced from 1.321 mg/100 mg wet tissue (control) to 0.811 mg/100 mg wet tissue when exposed at UVA and further reduced to 0.311 mg/100 mg wet tissue on UVB-irradiated fish. Therefore, the protein and lipid contents followed similar trends as carbohydrates (Table 3).

The ozone layer protects the Earth from the adverse effects of UV radiation. Currently, due to the development of industries and modern life, the expulsion of chlorofluorocarbons (CFCs) has increased exponentially. CFCs deplete the ozone layer, allowing massive amounts of UV radiation to impinge on the Earth's surface. As UV radiation reaches the Earth, it causes a series of serious problems for the ecosystem, especially living entities [25].

Since water covers the major part of the Earth's surface, the aquatic ecosystem encounters considerable adverse effects, and because fish are such important parts of both freshwater and marine ecosystems, there is much interest in studying them to improve survivability. Generally, during unfavourable or stressed conditions, fish require extra energy to restore normal physiological and biochemical changes for better survival. This excess energy needed may be obtained by oxidising the body's reserves, especially from carbohydrates and lipids [26]. The reduction in carbohydrate content in UVA- and UVB-radiation-exposed muscle tissues of freshwater fish *G. giuris* could be attributed to the use of stored carbohydrate, through aerobic glycolysis, to meet the extra energy demands during stress caused by UV radiation in the aquatic system [27]. When fish muscle tissue is exposed to radiation or human-made pollutions (e.g., industrial waste), these cause damage to the structure and morphology of the epidermis, degeneration of the epithelium, disappearance of striations, thickened septal wall, and broken fibres [21]. Similarly, the protein and lipid contents in the muscles tissue were also depleted or used due to the hypoxia in the surrounding environment. This study is useful to assess the quality of the environment, particularly with respect to global warming and environmental monitoring and impact assessment.

## 5. Conclusions

In the present study, short exposure times with ultraviolet rays UVA and UVB in muscle tissues of freshwater fish *Glossogobius giuris* demonstrated marked degeneration of the epithelium, the disappearance of striations, thickened septal walls, and broken fibres. The biochemical constituents of carbohydrate, protein, and lipid content showed a gradual reduction with the increase in wavelength of UVA and UVB. The ultrastructural and biochemical changes occurring depend on the dosage and exposal period of the UV radiation. Naturally, these results demonstrate that the forecast for Thandavarayankulam lake looks particularly bleak; this is supported by the fact that Solar radiation is on the increase in Tamil Nadu, as reported by various weather stations in the region.

**Author Contributions:** A.R.R. wrote the manuscript; K.K. prepared the images; P.A. prepared the Tables; M.S. and P.S. carried out experiments; S.J. edited the manuscript, edited images, and formatted the paper. All authors have read and agreed to the published version of the manuscript.

**Funding:** Internal funding was provided by the Department of Zoology, St. Xavier's College (autonomous), Palayamkottai, India.

**Institutional Review Board Statement:** Not applicable.

**Informed Consent Statement:** Not applicable.

**Acknowledgments:** We sincerely thank Sudhakar, Department of Biotechnology, M.S. University Tirunelveli, for suggestions on the fixation of tissues and microtome techniques. We also thank Ravi, Head of the Department, Department of Plant Science, M.S. University, Tirunelveli, for his constant support in microtome techniques and microscopy studies.

**Conflicts of Interest:** The authors declare no conflict of interest.

**Ethical Issue:** The fish were treated humanely according to MSU/DAS/EC/2016 (consultation 10 March 2017) and [12].

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
