# Peer review of "Deleterious Effect of Ultraviolet Radiation on Glossogobius giuris: A Short Experimental Study"

_sci, doi:10.3390/sci4010012_

Round 1

Reviewer 1 Report

The study is interesting since it deals with the effect of UV radiation on aquatic organism. I have marked some points that need revision, in order for the mansucript to be considered for publication. 

1) A general comment is how the water (sea or another ecosystem) conditions were reproduced in the lab. I miss the details of this part that should be added in more detail. This comment is connected also to page 8, lines 193-197, in which you refer to Sun UV radiation strength (this lines need revision, are difficult to follow). This gap in the simulation of pragmatic conditions should be addressed. 

2) Please check again the sections of the manuscript. For example, is section 2 separate, or should be somehow incorporated in "material and methodologies"?

3) For ethical issues, please consult the paper (and info inside) 

Bennett, R.H., Ellender, B.R., Mäkinen, T., Miya, T., Pattrick, P., Wasserman, R.J. et al., 2016, ‘Ethical considerations for field research on fishes’, Koedoe 58(1), a1353. http:// dx.doi.org/10.4102/koedoe. v58i1.1353 

4) Minor edits: Page 2, line 64 is "classified". Page 4, line 115, please substitute litre with L. Also in the same paragraph check the Celsius symbol. 

5)  Please revise Table and Figure legends. In some of them, the beginning is "explanatory", e.g., Table 2. summarizes .... In addition, please consider the title of section 4.2. (i miss the carbohydrates there). 

Author Response

Reviewer 1

The study is interesting since it deals with the effect of UV radiation on aquatic organism. I have marked some points that need revision, in order for the manuscript to be considered for publication.

SJ: Note to the reviewer. Requested corrections and amendments are highlighted yellow on the MS.

1) A general comment is how the water (sea or another ecosystem) conditions were reproduced in the lab. I miss the details of this part that should be added in more detail. This comment is connected also to page 8, lines 193-197, in which you refer to Sun UV radiation strength (these lines need revision, are difficult to follow). This gap in the simulation of pragmatic conditions should be addressed. SJ: this is already addressed in lines 116 to 122 in the MS.

2) Please check again the sections of the manuscript. For example, is section 2 separate, or should be somehow incorporated in "material and methodologies"? SJ: No, the authors believe it stays as separate section because we are setting out motivation for these experiments in clear terms.

3) For ethical issues, please consult the paper (and info inside)

Bennett, R.H., Ellender, B.R., Mäkinen, T., Miya, T., Pattrick, P., Wasserman, R.J. et al., 2016, ‘Ethical considerations for field research on fishes’, Koedoe 58(1), a1353. http:// dx.doi.org/10.4102/koedoe. v58i1.1353. SJ: thanks, the reviewer for his/her assistance and recommendation. In addition to the recommended paper, we have already stated that the fish were treated humanely according to Manonmaniam Sundaranar University guidelines entitled: MSU/DAS/EC/2016 (consultation 10/03/2017) as well as the following website: https://www.understandinganimalresearch.org.uk/openness/regulation/. The recommended paper has been cited in the MS.

4) Minor edits: Page 2, line 64 is "classified"—SJ: corrected. Page 4, line 115, please substitute litre with L SJ: corrected. Also, in the same paragraph check the Celsius symbol SJ: corrected.

5)  Please revise Table and Figure legends. In some of them, the beginning is "explanatory", e.g., Table 2. summarizes .... In addition, please consider the title of section 4.2. (i miss the carbohydrates there)—SJ: all Figures and Tables have been revisited and revamped

Reviewer 2 Report

In the study” Deleterious Effect of Ultraviolet Radiation on Glossogobius 2 giuris: A Short Experimental Study”, authors described the effects of UVA or UVB rays on fish muscle. Solar ultraviolet radiation, especially at the higher doses, can have deleterious effects on living organisms exposed to sun. The water ecosystem is directly affected which can further make a difference on a food chain, including fish. Therefore, testing the direct effects of UV rays on fish can be very beneficial. The authors demonstrated marked degeneration in muscle tissue of the fish exposed to UVA or UVB under laboratory conditions.

However, the authors did not properly explain why the study is performed on the muscle tissue, when epidermis and dermis serve as a barrier for UV rays. In addition, how do you explain the appearance of pigmentations in muscles, but not in epidermis? The study is useful for collecting more data related to global warming situation, but data presented should be better clarified.

Author Response

REVIEWER 2

SJ: Note to the reviewer. Requested corrections and amendments are highlighted yellow on the MS.

In the study” Deleterious Effect of Ultraviolet Radiation on Glossogobius giuris: A Short Experimental Study”, authors described the effects of UVA or UVB rays on fish muscle. Solar ultraviolet radiation, especially at the higher doses, can have deleterious effects on living organisms exposed to sun. The water ecosystem is directly affected which can further make a difference on a food chain, including fish. Therefore, testing the direct effects of UV rays on fish can be very beneficial. The authors demonstrated marked degeneration in muscle tissue of the fish exposed to UVA or UVB under laboratory conditions.

However, the authors did not properly explain why the study is performed on the muscle tissue, when epidermis and dermis serve as a barrier for UV rays. SJ: it is true epidermis and dermis can offer some barrier to UV rays. However, when fish are swimming near the surface of the water, this is when UV rays do the most damage. Such damage could persist from spawning of eggs through to adulthood. This has been explained in the MS. In addition, how do you explain the appearance of pigmentations in muscles, but not in epidermis? SJ: in our experiment we have not seen any changes in appearance within the epidermis. Further, we have just concentrated on the presence of melanocyte cells in muscles (stratum laxum, stratum compactum) which is indicative of the appearance of pigmentations in muscles. The study is useful for collecting more data related to global warming situation, but data presented should be better clarified. SJ: on page 9 of the MS, beginning with line 222 through to 245, we have addressed climatic conditions thoroughly.

Round 2

Reviewer 1 Report

The authors addressed the majority of comments. The manuscript cn be accepted for publication.